# Fine Feature Reconstruction in Point Clouds by Adversarial Domain Translation

Prashant Raina*        Tiberiu Popa        Sudhir Mudur

Department of Computer Science and Software Engineering
Concordia University

## ABSTRACT

Point cloud neighborhoods are unstructured and often lacking in fine details, particularly when the original surface is sparsely sampled. This has motivated the development of methods for reconstructing these fine geometric features before the point cloud is converted into a mesh, usually by some form of upsampling of the point cloud. We present a novel data-driven approach to reconstructing fine details of the underlying surfaces of point clouds at the local neighborhood level, along with normals and locations of edges. This is achieved by an innovative application of recent advances in domain translation using GANs. We "translate" local neighborhoods between two domains: point cloud neighborhoods and triangular mesh neighborhoods. This allows us to obtain some of the benefits of meshes at training time, while still dealing with point clouds at the time of evaluation. By resampling the translated neighborhood, we can obtain a denser point cloud equipped with normals that allows the underlying surface to be easily reconstructed as a mesh. Our reconstructed meshes preserve fine details of the original surface better than the state of the art in point cloud upsampling techniques, even at different input resolutions. In addition, the trained GAN can generalize to operate on low resolution point clouds even without being explicitly trained on low-resolution data. We also give an example demonstrating that the same domain translation approach we use for reconstructing local neighborhood geometry can also be used to estimate a scalar field at the newly generated points, thus reducing the need for expensive recomputation of the scalar field on the dense point cloud.

**Index Terms:** Computing methodologies—Computer graphics—Shape modeling—Point-based models; Computing methodologies—Machine learning—Machine learning approaches—Neural networks

## 1 INTRODUCTION

Point clouds and meshes are two representations of 3D surfaces that have long coexisted in the fields of computer graphics and computer vision. Point clouds are easier to acquire from the real world. However, they lack most of the geometric information which makes 3D meshes indispensable. The connectivity information provided by meshes allows one to easily calculate normals, curvatures and other geometric properties of the underlying surface. They can also be remeshed, or resampled to arbitrary precision. This gap between point clouds and meshes has traditionally been bridged by fitting parametric or implicit surfaces to point cloud neighborhoods. However, recent advances in deep learning have now made it possible to propose data-driven approaches to transforming data between these two domains.

We present here a simple and elegant approach to reconstructing fine features of surfaces sampled as point clouds. By "fine features", we refer specifically to features smaller than the separation between

---

*e-mail: prashantraina2005@gmail.com

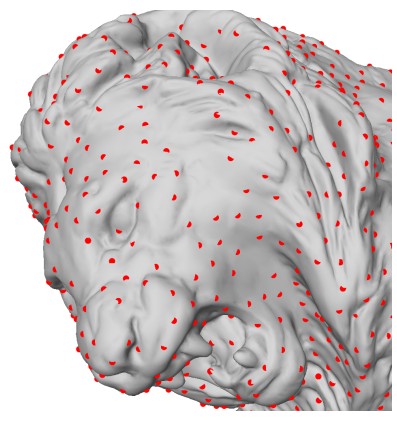

Figure 1: A sparse point cloud (red points) sampled from a detailed surface. Fine features such as the eye that lie between the sampled points are challenging to reconstruct.

sampled points (an example would be the eye in Figure 1). Naturally, it is impossible to recover all fine features in the general case, as undersampling causes information to be lost. However, the field of deep learning enables us to train a machine learning model on a large dataset and obtain a learned prior which will allow us to "hallucinate" fine details that would be expected from the underlying distribution of the dataset.

Our approach leverages generative adversarial networks (GANs) for *domain translation*. GANs are a class of deep neural networks which has shown great potential in synthesizing realistic novel images that appear to be sampled from a particular domain of image data. One variant of the GAN architecture which is particularly interesting to us is the conditional GAN architecture. Conditional GANs have been successfully applied to domain translation, i.e. transforming images between two very different but related domains [11]. Some examples of domain translation include translating sketches of handbags to photographs of handbags, street maps to satellite images, or summer landscape photographs to winter landscape photographs. We tackle the fine feature reconstruction problem at the local neighborhood level, by framing it as a domain translation problem between two kinds of local heightmaps: "sparse" heightmaps, which are sampled from point clouds, and "dense" heightmaps, which are sampled from meshes using raycasting. This is a very unique and atypical domain translation problem because quantitative accuracy is extremely important. By contrast, the results in the aforementioned traditional domain translation problems need only appear qualitatively plausible.

Our main contribution is in adapting existing work on domain translation to the problem of reconstructing fine features from low-resolution point clouds. Our feature reconstruction results are superior to the state-of-the-art methods [22, 28]. These methods use a completely different patch-based approach along with much more complex neural network architectures. Our method runs in significantly less time than the most recent state-of-the-art method [22].

Furthermore, our method generalizes well to low-resolution point clouds, even when all the training inputs are sampled from high-resolution point clouds. This highlights the robustness of our domain translation approach. The implications of this method go beyond point positions, and we show that it can easily be extended to interpolate values of a scalar field at the newly created points in the dense point cloud. As a bonus, we can also easily obtain normals at the points in a dense heightmap.

Our paper is organized as follows: Section 2 recaps the most relevant related work in point cloud consolidation, as well as domain translation. Section 3 describes in detail how we generate the two types of heightmaps and perform domain translation. Section 4 shows the surface reconstruction results we obtain from applying heightmap domain translation to point cloud neighborhoods. A detailed quantitative evaluation of the results and comparison with previous methods is given in Section 5. We briefly describe an application to upsampling of a scalar field in Section 6, before giving our conclusions in Section 7. There are also two appendices, A and B, which give neural network training details and additional figures of results.

## 2 RELATED WORK

Image-to-image translation using deep learning has gained a lot of attention since Isola *et al* published their seminal work [11] on using conditional adversarial networks for translating between image domains given a training set of paired examples. This quickly led to more work on unpaired image translation [31], as well as image translation between more than two domains [2]. There have also been attempts to bring GAN-based domain translation methods to other domains such as natural language text [27], audio [9] and voxel-based 3D data [4]. Some recent work on point clouds such as FoldingNet [26] and AtlasNet [6] can also, in some sense, be regarded as precursors to domain translation between point clouds and explicit surfaces.

Reconstructing fine features from point clouds can take different forms depending on how the point cloud is sampled. In some cases, the points are essentially pixels obtained from a depth image [10,23], which means that the points have a 2D grid structure that can be treated as a Euclidean domain for discrete convolution. Furthermore, depth images are almost invariably accompanied by color or intensity images, which provide vital information that is exploited in past work. In our work, we focus on the more general problem of reconstructing detailed surfaces from completely unstructured point clouds by first increasing the density of the point clouds before attempting to reconstruct the surface. This is most closely related to the problem of point cloud *consolidation*, where the goal is to obtain a point cloud representation from which an accurate 3D mesh can be reconstructed [7]. A variety of procedural point cloud consolidation methods have been proposed, including LOP [15], WLOP [7], EAR [8] and deep points consolidation [24]. All of these methods involve fitting local geometry to point clouds, and WLOP and EAR have been incorporated into popular geometry processing libraries.

Recent years have seen a wave of interest in applying deep learning to point clouds, sparked by the success of PointNet [16] and its multi-scale variant, PointNet++ [17], in the field of point cloud classification and semantic segmentation. This has led to point cloud consolidation [20] and upsampling [30] approaches based on PointNet, as well as a family of point cloud upsampling techniques based on PointNet++ that includes PU-Net [29], EC-Net [28] and 3PU [22]. Li *et al* have concurrently developed a GAN-based point cloud upsampling method [14] which uses generator and discriminator architectures loosely based on PU-Net and 3PU. By contrast, our work uses local heightmaps, much like Roveri *et al* [20]. However, Roveri *et al* focus on consolidation of already dense point clouds with 50 thousand points, while we are mainly interested in reconstructing fine features from much sparser point clouds (5000 points

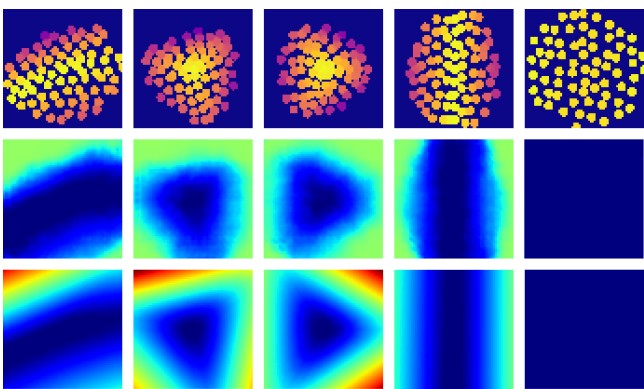

Figure 2: Heightmaps sampled from the *fandisk* model. The sparse and dense heightmaps have different color maps to reflect the fact that sparse heightmaps are not occupied at all pixels. Top row: sparse heightmaps obtained from the point cloud. Middle row: dense heightmaps predicted by our GAN. Bottom row: ground truth heightmaps obtained by casting rays onto the original mesh.

or less). Moreover, they do not use domain translation, since their ground truth data is also obtained from point clouds. We compare our work to the publicly available implementations of EC-Net [28] and 3PU [22], since they are the most recently published state-of-the-art works. 3PU is claimed to have superior results to all previous point cloud upsampling work.

## 3 HEIGHTMAP DOMAIN TRANSLATION

The key idea behind our method is that reconstructing fine features of point cloud neighborhoods can be reduced to an image-to-image translation problem between two kinds of local heightmaps (examples shown in Figure 2):

1. Sparse heightmaps, which are sampled from point clouds. By "sparse", we mean that not all pixels in the heightmap are occupied. This follows naturally from the fact that point clouds have gaps between points. In practice, we also set an upper limit on the number of points contributing to a sparse heightmap, which makes our method more robust while also speeding up computation of the sparse heightmaps. Therefore, they are also sparse in the conventional sense that there are $O(1)$ points represented in the heightmap.

2. Dense heightmaps, which are sampled from meshes. Since the mesh is typically defined in a continuous manner over a neighborhood, we can obtain a heightmap by casting rays onto the mesh. It is worth noting that this raycasting approach can easily be modified to obtain other kinds of local "maps", such as scalar or vector fields defined over the mesh.

In this section, we first explain how these two types of heightmaps are computed (Section 3.1). We then describe how we translate heightmaps from sparse to dense, using a well-known image-to-image translation approach that exploits a conditional GAN (Section 3.2). We finally show how our overall approach has additional benefits for estimating normals (Section 3.3).

### 3.1 Local Heightmap Computation

Some aspects of heightmap computation are common to both the sparse and dense local heightmaps. The heightmap is assigned a size of $k \times k$ pixels, where we generally select $k = 64$. The side of the heightmap corresponds to a length of $2r$ in the space of the input shape, where $r$ is the *search radius* used to collect nearest neighbors from the point cloud. The local coordinate frame of the

neighborhood is centered on a particular point of the point cloud, which has a corresponding oriented normal **n**. The horizontal and vertical directions of the local frame are given by an arbitrary tangent **t** and its corresponding bitangent $\mathbf{b} = \mathbf{n} \times \mathbf{t}$. In our experiments, the tangents are randomly rotated in the tangent plane at both training and testing time, so that there is no bias introduced by the choice of tangent direction.

### 3.1.1 Sparse Heightmap Computation

For computing sparse heightmaps from point clouds, we adapt a simple and efficient representation used in earlier works [19, 20]. A random set of neighboring points (limited to 100) is chosen from within the search radius $r$. Since our sampled points have consistent associated normals, we can omit neighbors with back-facing normals. The neighborhood is scaled by a factor of $1/r$ to reduce dependence on scale. These points are then projected orthogonally onto the local tangent plane, i.e. the plane of the heightmap image. For each pixel in the heightmap image, we can easily compute the corresponding pixel center in the local coordinate frame of the neighborhood. We can then compute the intensity of each pixel as the weighted average of the signed distances of nearby projected points from their original positions. The unnormalized weights have a Gaussian falloff $w_i = exp(-\frac{d_i^2}{2\sigma^2})$, where $d_i$ are the distances of the projected points from the pixel center in the image plane (we set $\sigma = 5r/k$). A constant value 1 is added to all projection heights, so that the value 0 is reserved for unoccupied pixels. As shown in section 6, the same approach can also be used to generate a sparse map, not for height, but for a scalar field.

### 3.1.2 Dense Heightmap Computation

Given a point on a surface represented by a mesh (which need not be a vertex), and its corresponding normal, we can use raycasting to generate a dense heightmap. We first transform the center of each heightmap pixel to the space of the mesh, using the local tangent frame mentioned earlier. From each pixel center, we shoot two rays in opposite directions perpendicular to the tangent plane. If both rays intersect the mesh, we chose the nearer intersection point. The intensity of the pixel is the signed distance of the pixel center to the intersection point, or a fixed large value (10) in the event that neither ray intersects the mesh. These raycasting operations are performed efficiently using the Embree raycasting framework [21], which also provides us with the intersected triangle ID and the barycentric coordinates of the intersection point. This additional information can be used to interpolate scalar or vector fields previously computed on the vertices of the mesh, in order to generate other kinds of dense local maps.

### 3.2 Image-to-Image Translation

We use a conditional generative adversarial network (cGAN) to perform image-to-image translation between our two heightmap domains, following the approach by Isola *et al* [11]. The outputs of both the generator $G(x, z)$ and discriminator $D(x, y)$ are conditioned on the input image $x$. The discriminator does not attempt to classify the entire input image as real or fake, but rather classifies individual patches ($8 \times 8$ in the case of our dense heightmaps). The generator must minimize two losses:

1. The GAN loss, $\mathbb{E}_{x,y}[D(x,y)] + \mathbb{E}_{x,z}[1 - D(G(x,z))]$, which preserves high-frequency similarities between $G(x, z)$ and the corresponding ground-truth image $y$.

2. The $\ell_1$ loss, $\mathbb{E}_{x,y,z}|y - G(x,z)|_1$, which preserves low-frequency similarities with the ground truth.

In practice, the noise vector $z$ is introduced implicitly by random dropout of neurons with 50% probability. It is worth noting that the sparse heightmaps we use contain incomplete information which

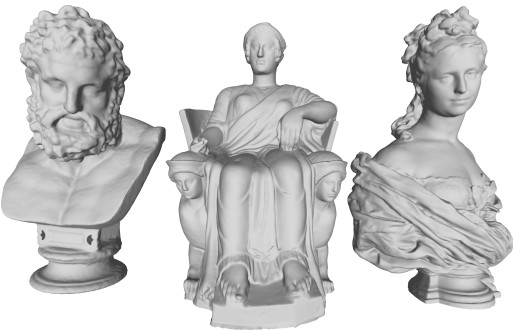

Figure 3: Examples of training models from SketchFab.

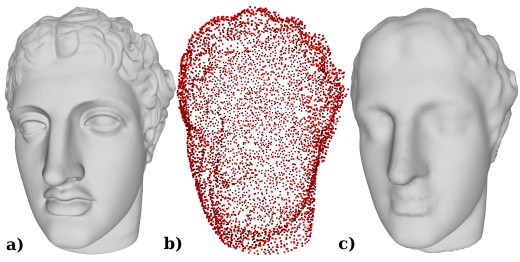

a)     b)     c)

Figure 4: a) Ground truth *head* model. b) 5000 points sampled to act as testing input. c) Poisson reconstruction of the input point cloud, showing the loss of features due to undersampling.

could imply multiple different dense heightmaps. We therefore have the option of enabling neuron dropout at the time of evaluating the network, in order to obtain a stochastic output.

### 3.3 Prediction of Normals

The regular grid structure of a heightmap provides us the additional benefit that it allows easy computation of normals at each point on the heightmap. Given a dense heightmap, we can use backward differences to estimate gradients in the tangent and bitangent directions. This gives us the approximate normal at each point as the direction of the vector $(\frac{\partial h}{\partial x}, \frac{\partial h}{\partial y}, \frac{2r}{k})$. This approximate normal map proved to be sufficient for our purposes, although it could also be refined using a neural network.

## 4 FINE FEATURE RECONSTRUCTION

We trained our GAN on local neighborhoods sampled from a set of 90 meshes of statues obtained from SketchFab. The meshes are identical to the training set used by Wang *et al* [22] for training 3PU. These meshes were generated by 3D-scanning statues of people and animals (Figure 3). Training details such as hyperparameters are given in Appendix A. For evaluation, we used a separate set of 16 meshes of statues from SketchFab. These include all 13 testing meshes used by Wang *et al* [22], as well as 3 additional meshes that we procured. All ground truth meshes have several hundred thousand vertices.

To obtain a sparse set of points for reconstructing fine geometric features, we randomly sample a fixed number of points from a test mesh using Poisson disk sampling (using the implementation available in the VCG library [1]). Figure 4 shows an example of sampling these points. Normals for these points are estimated using PCA, based on 30 nearest neighbors from the sparse point cloud.

After generating $64 \times 64$ sparse heightmaps for these sampled points using the method described in section 3.1.1, we use our trained model to suggest a plausible $64 \times 64$ dense heightmap. Given

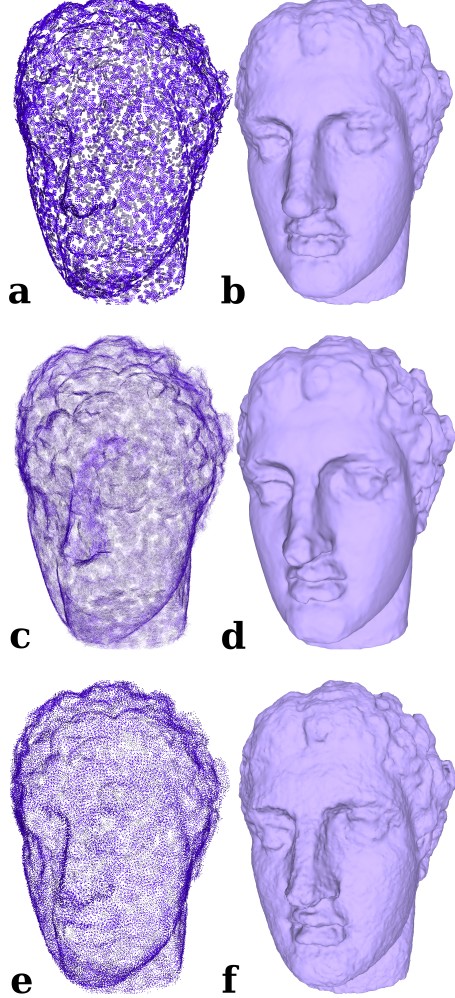

Figure 5: Our domain translation and reconstruction results for the 5000 points in Figure 4.b, using different modes. a, c and e correspond to the *detailed*, *superdense* and *even* modes listed in Section 4. b, d and f are their respective reconstructions using screened Poisson surface reconstruction.

the search radius as well as the normal and tangent vectors used to obtain the sparse heightmap, we can easily transform pixels of the dense heightmap back into points in the space of the original point cloud. Rather than converting all 4096 pixels into points, we take points from an $8 \times 8$ or $16 \times 16$ square in the middle of the generated heightmap. The rationale for this is that the conditional GAN is transforming heightmaps at the local level with no global information, and therefore we cannot expect the extremities of the newly generated heightmap to be accurate, even if it appears to be a plausible translation of the given sparse heightmap. Once normals are computed using the method in Section 3.3, the surface can then be easily reconstructed using a conventional surface reconstruction algorithm. In our case, we use screened Poisson reconstruction [12].

We experimented with different combinations of heightmap search radius, size of the central square, and stride (a stride of 2 implies taking every alternate row and column). Note that $\delta$ is the median distance between a point and its nearest neighbor in the input point cloud, which we use as a measure of scale. Out of these combinations, we found three to be interesting:

1. *detailed* mode: Radius $4\delta$, central $8 \times 8$, stride 2 (16x upsampling). This mode produces the most detailed looking mesh

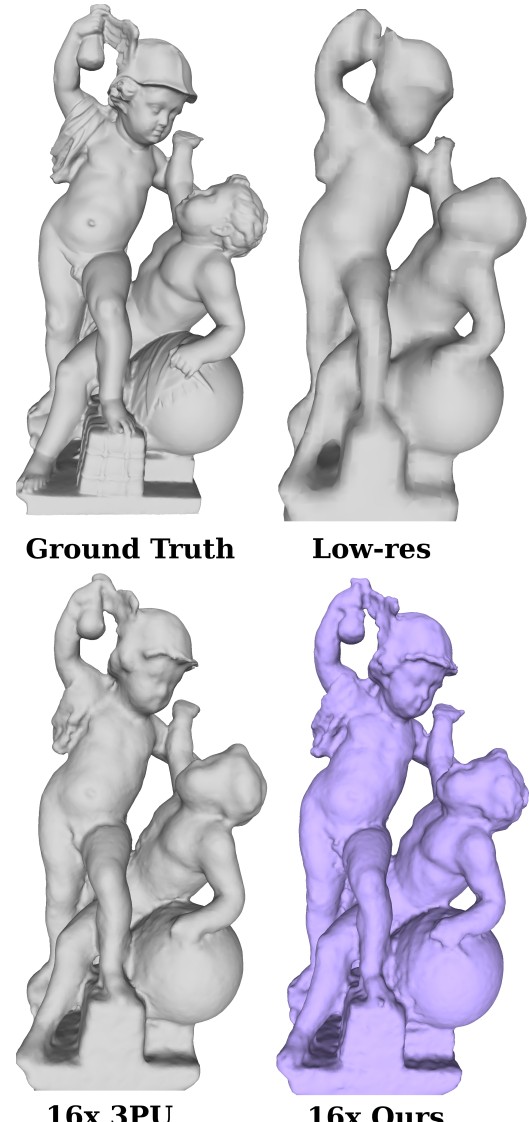

**Ground Truth**          **Low-res**

**16x 3PU**          **16x Ours**

Figure 6: Results for the statue *Cupid Fighting*, reconstructed from a sample of 5000 points from the ground truth mesh.

when screened Poisson reconstruction is applied.

2. *superdense* mode: Radius $4\delta$, central $16 \times 16$, stride 1 (256x upsampling). This mode produces an extremely dense point cloud, from which a reasonably accurate mesh can be reconstructed.

3. *even* mode: Radius $8\delta$, central $8 \times 8$, stride 2 (16x upsampling). The high-resolution point cloud obtained with this mode appears to be the most evenly sampled.

Figure 5 compares the results for the three modes, after processing the sampled points from Figure 4.b.

### 4.1  Reconstruction Results

Examples of our surface reconstruction results are shown in Figures 6 to 8. Note that there were no post-processing operations such as smoothing in any of our figures. In Figures 6 and 7, we can see that our method is able to reconstruct fine details such as facial features from a sample of only 5000 points. In Figure 7, we can see that the

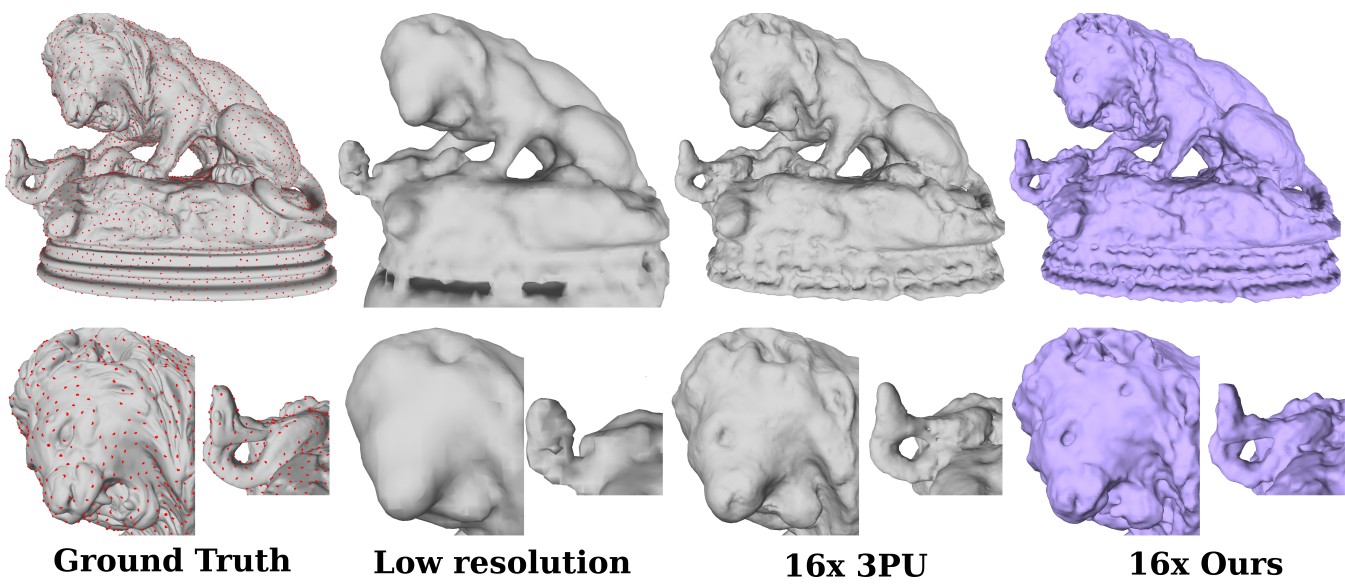

**Ground Truth**  **Low resolution**  **16x 3PU**  **16x Ours**

Figure 7: Reconstruction results for the statue *Lion Étouffant Un Serpent*. The 5000 input points are overlaid on the ground truth mesh.

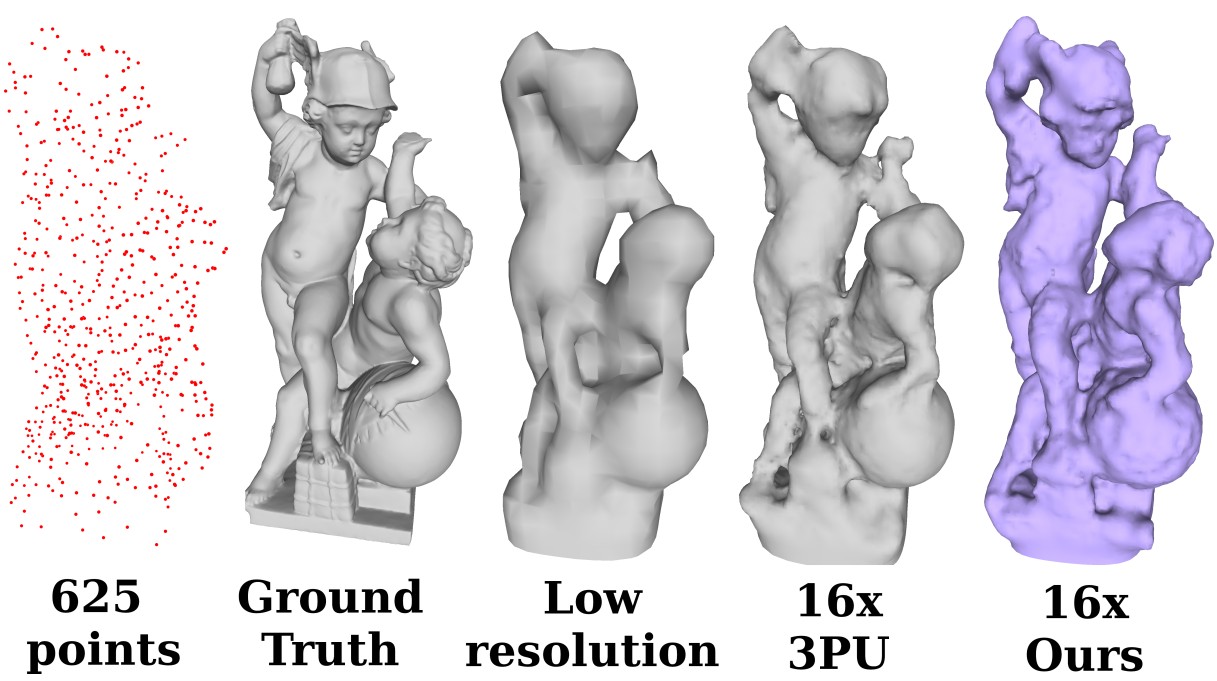

**625 points**  **Ground Truth**  **Low resolution**  **16x 3PU**  **16x Ours**

Figure 8: Reconstruction results for an extremely low-resolution sample of 625 points (in red) from *Cupid Fighting*. We compare against the best results we were able to obtain for 3PU, across multiple random samplings of 625 points.

eye of the lion from Figure 1 has been reconstructed, along with other fine features such as the teeth. Figure 8 shows an extreme example where we upsample a set of only 625 points, while still being able to reconstruct features such as the wings on the helmet. For comparison, we show the corresponding result using the latest state-of-the-art method by Wang *et al* [22], informally called 3PU. This method has already been shown to be superior to state-of-the-art approaches such as EC-Net [28] and PU-Net [29]. Additional results can be found in Appendix B.

During our experiments on extremely low resolution point clouds, we naturally found that different random samplings of 625 input points for the same testing mesh give slightly different resulting meshes. We found that our method produced fairly consistent results across different random samplings. This variation is much larger for the PointNet++-based approaches. Therefore, we have selected the best result we obtained for 3PU [22] to compare with our result in Figure 8. Figure 14 in Appendix B compares our results with 3PU for multiple different random samplings of 625 input points.

## 5 EVALUATION

We evaluate our surface reconstruction results quantitatively based on three metrics:

1. Distance of each point in the high-resolution point cloud to the ground truth mesh (D2M).

2. Hausdorff distance (HD): the maximum distance between a point on the reconstructed mesh and its nearest neighbor on the ground truth mesh:

$$\max\left(\max_{p\in P}\min_{q\in Q}\|p-q\|^2, \max_{q\in Q}\min_{p\in P}\|p-q\|^2\right) \quad (1)$$

3. Chamfer distance (CD) [5]: the mean distance between a point on the reconstructed mesh and its nearest neighbor on the ground truth mesh:

$$\frac{1}{2}\left(\frac{1}{|P|}\sum_{p\in P}\min_{q\in Q}\|p-q\|^2 + \frac{1}{|Q|}\sum_{q\in Q}\min_{p\in P}\|p-q\|^2\right) \quad (2)$$

We used Meshlab [3] to automate computation of all three metrics. The Hausdorff distance and Chamfer distance can be computed simultaneously by first randomly sampling a large number of points from the vertices, edges and faces of one mesh. The number of points sampled is equal to the number of vertices on the sampled mesh (several hundred thousand). We then find the mean and maximum distances between these points and the nearest vertices on the other mesh. Pairs of points are discarded if their separation is greater than 5% of the diagonal of the bounding box. The above procedure is then repeated in the opposite direction, giving us two maximums and two means. The Hausdorff distance is the maximum of the two maximums and the Chamfer distance is the mean of the two means. Note that for performing screened Poisson reconstruction on point clouds upsampled using EC-Net and 3PU, we estimate normals using PCA based on 30 nearest neighbors. This is not necessary for our method, since we obtain normals from the dense heightmap as mentioned in Section 3.3.

In Table 1, we compare the results for the three modes for sampling our dense heightmaps, as defined in section 4.1. To have a sense of the scale, note that the point coordinates were normalized to lie within a unit cube. We first consider the *detailed* and *even* modes, which both produce 16 times the number of input points. Although the *detailed* mode produces the best reconstructed mesh, it is noteworthy that the *even* mode is not far behind, and even surpasses the *detailed* mode for very low resolution point clouds. Those seeking to apply our method to obtain a point cloud, and not a mesh, have

| Mode | D2M | HD | CD |
|---|---|---|---|
| Detailed – 5000 points | **3.42E-04** | **3.24E-02** | **1.03E-03** |
| Even – 5000 points | 3.96E-04 | 3.48E-02 | 1.42E-03 |
| Superdense – 5000 points | 5.89E-04 | 3.35E-02 | 1.37E-03 |
| Detailed – 2500 points | **4.00E-04** | **3.66E-02** | **1.53E-03** |
| Even – 2500 points | 4.38E-04 | 4.08E-02 | 2.11E-03 |
| Superdense – 2500 points | 8.39E-04 | 4.00E-02 | 2.25E-03 |
| Detailed – 625 points | 1.16E-03 | 5.20E-02 | **3.86E-03** |
| Even – 625 points | **1.10E-03** | **5.03E-02** | 5.25E-03 |
| Superdense – 625 points | 2.93E-03 | 5.81E-02 | 6.23E-03 |

Table 1: Quantitative comparison of reconstruction results using our three modes for domain translation of input point clouds with 5000, 2500 and 625 points. Note that the *superdense* mode produces 256 times the number of points, while the others increase the density by a factor of 16.

| Size of point clouds | D2M | HD | CD |
|---|---|---|---|
| >300K: 5000 points | **3.42E-04** | **3.24E-02** | **1.03E-03** |
| 5K: 5000 points | 8.04E-04 | 3.26E-02 | 1.30E-03 |
| 625: 5000 points | 1.17E-03 | 3.38E-02 | 1.54E-03 |
| >300K: 2500 points | **4.00E-04** | **3.66E-02** | **1.53E-03** |
| 5K: 2500 points | 9.63E-04 | 3.72E-02 | 1.90E-03 |
| 625: 2500 points | 1.46E-03 | 3.96E-02 | 2.17E-03 |
| >300K: 625 points | 1.16E-03 | 5.20E-02 | **3.86E-03** |
| 5K: 625 points | **1.09E-03** | **5.04E-02** | 4.32E-03 |
| 625: 625 points | 1.97E-03 | 5.54E-02 | 4.58E-03 |

Table 2: Quantitative comparison of using different point cloud sizes when training: all mesh vertices (over 300K points), 5000 points sampled using Poisson disk sampling, or 625 points. The resulting metrics are compared for input point clouds with 5000, 2500 and 625 points.

| Method | D2M | HD | CD |
|---|---|---|---|
| EC-Net – 5000 points | 3.42E-04 | 6.30E-02 | 3.89E-03 |
| 3PU – 5000 points | **2.91E-04** | 3.63E-02 | 1.32E-03 |
| Ours – 5000 points | 3.42E-04 | **3.24E-02** | **1.03E-03** |
| EC-Net – 2500 points | 6.56E-04 | 6.57E-02 | 5.80E-03 |
| 3PU – 2500 points | **3.16E-04** | 4.86E-02 | 2.13E-03 |
| Ours – 2500 points | 4.00E-04 | **3.66E-02** | **1.53E-03** |
| EC-Net – 625 points | 1.85E-03 | 5.84E-02 | 8.12E-03 |
| 3PU – 625 points | 1.31E-03 | 5.55E-02 | 4.96E-03 |
| Ours – 625 points | **1.16E-03** | **5.20E-02** | **3.86E-03** |

Table 3: Quantitative comparison of reconstructed surfaces after 16x upsampling using EC-Net, 3PU and our *detailed* mode for point clouds with 5000, 2500 and 625 points.

| Method | D2M | HD | CD |
|---|---|---|---|
| 3PU – 5000 points | 0.081 | 1.83 | 0.440 |
| Ours, Detailed – 5000 points | **0.066** | **1.26** | **0.426** |
| Ours, Even – 5000 points | 0.113 | 1.36 | 0.443 |
| 3PU – 2500 points | 0.161 | 2.77 | 0.485 |
| Ours, Detailed – 2500 points | **0.112** | **1.71** | **0.442** |
| Ours, Even – 2500 points | 0.181 | 1.88 | 0.478 |
| 3PU – 625 points | 0.593 | 8.13 | 0.902 |
| Ours, Detailed – 625 points | **0.325** | **3.48** | **0.599** |
| Ours, Even – 625 points | 0.492 | 4.03 | 0.756 |

Table 4: Quantitative comparison of reconstructed surfaces after 16x upsampling using 3PU, our *detailed* mode and our *even* mode on the ABC dataset [13], for point clouds with 5000, 2500 and 625 points.

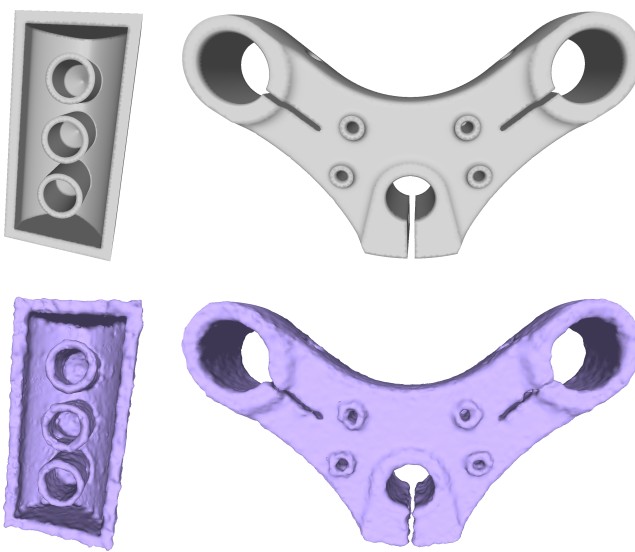

Figure 9: Two examples of reconstruction results obtained on the ABC dataset [13], after upsampling samples of 2500 points. Top row: ground truth. Bottom row: our results using the *detailed* mode.

a choice between using our *even* mode, or alternatively performing Poisson disk sampling on the mesh produced using our *detailed* mode. The *superdense* mode, which produces 256 times the number of input points, suffers a bit when it comes to the distance to the ground truth mesh. However, it is not far behind the other modes when it comes to the Hausdorff and Chamfer distances.

A unique feature of our method is that we can obtain good results on low resolution point clouds, even after training on high-resolution point clouds. Our training meshes are very dense, and each has over 300 thousand vertices. They can also be randomly downsampled to a lower resolution such as 5000 or 625 points during training. We have therefore investigated how different resolutions of the training point clouds affect the surface reconstruction results. After all, the resolution of the training point clouds does affect the variety of the sparse heightmaps that will be seen during training. Table 2 shows that using the entire set of mesh vertices during training usually produces the best results. Even in the case where the testing point clouds have 625 points, it is a GAN trained on a higher resolution (5000 points) which gives the best results. This is likely due to a combination of two factors: i) we use raycasting onto meshes to obtain our ground truth dense heightmaps, and ii) we randomly sample only 100 points from each local neigborhood to contribute to the sparse heightmap, thereby making our domain translation more robust.

Table 3 compares the accuracy of the meshes produced by applying screened Poisson reconstruction on the dense output point cloud of our domain translation method, as well as other recent methods for point cloud upsampling. We choose the *detailed* mode of domain translation for comparison, since it upsamples point clouds by 16 times (thereby allowing a fair comparison with 3PU), and it also produces the most accurate reconstructed meshes. We compare our work with the results obtained using EC-Net [28], as well as the recent state-of-the-art method called 3PU by Wang *et al* [22]. Wang *et al* claim that 3PU gives better results than all previous work for 16x upsampling of point clouds. In order to upsample point clouds 16x using EC-Net, we apply 4x upsampling with EC-Net twice in succession, as recommended by Yu *et al* to Wang *et al* [22]. As mentioned earlier, we use the same training data as Wang *et al*. We do not retrain EC-Net, as we do not have edge annotations in our training data, which are required by EC-Net. From Table 3, we can

see that across all three metrics, our quantative results are clearly superior to those of EC-Net. Our results are also superior to 3PU for the Hausdorff distance and Chamfer distance metrics, while still being competitive for the D2M metric.

We have additionally performed a large-scale quantative evaluation of our method on the first 1,000 OBJ files in the ABC dataset [13], whose results are summarized in Table 4. Figure 9 shows examples of our results on this dataset. We did not re-train our GAN or the 3PU network on this dataset, but rather used the same networks which were already trained on the aforementioned 90 models from SketchFab. The 3D models in the ABC dataset are densely sampled triangular meshes obtained from parametric CAD models. For models containing multiple connected components, only the largest connected component was used for the evaluation. The coordinates of these models are not normalized. Therefore, in order to make a fair aggregation of the results, we divided the computed distance metrics for each model by the average distance between pairs of neighboring mesh vertices. We also compute the median of each metric over all 1,000 models, to reduce the effect of outlier cases where the point clouds produced by 3PU do not work well with the normal estimation method (there were no such cases among the SketchFab testing models). The results show that our method gives reconstructed surfaces that are more accurate than those obtained using 3PU. Furthermore, the worst-case error given by the Hausdorff distance is comparable to the normalized distance of 1 between nearest neighbors in the ground-truth dense mesh, while the average-case errors are significantly smaller than 1.

We also found our method to be extremely fast, taking only 3 minutes to upsample a large point cloud of 160K points with an unoptimized implementation. By contrast, the far more complicated neural network of 3PU took 214 minutes to perform the same task using the code provided by the authors. We note, however, that both methods have comparable speed for small point clouds. For instance, they both take around 20 seconds to upsample 5000 points.

The aforementioned state-of-the-art point cloud upsampling methods are ultimately based on PointNet++ [17]. It is therefore worth noting that all upsampling methods based on PointNet++ share certain weaknesses:

1. They are not invariant to permutations of the point cloud. This is because they all rely on farthest-point sampling as an initial step for their neural networks (see [25]). Our sparse heightmaps, on the other hand, are completely invariant to permutations of the points in the local neighborhood.

2. They rely on large and complicated neural network architectures, which affects the computational efficiency. By contrast, we use a simple convolutional U-Net architecture for the GAN.

3. They do not perform well in regions with unusually dense sampling. This is because after the farthest-point sampling step in each set abstraction layer, K-nearest neighbors are taken as representatives of the local neighborhood. Therefore, regions of high density will result in too many neighbors that are very close to the center point, thus giving a skewed picture of the local neighborhood. In our case, having many points mapping to the same pixels of the sparse heightmap will not significantly affect the pixel intensities.

## 6 SCALAR FIELD UPSAMPLING

Our method also has the potential for using the conditional GAN to predict values of a scalar or vector field corresponding to the dense heightmap at a given point. As an example, we show an application to the *sharpness field* defined by Raina *et al* [18, 19]. The feature-aware smoothing method described in [19] is mainly concerned with the local maxima of the field, which are scale-invariant. This

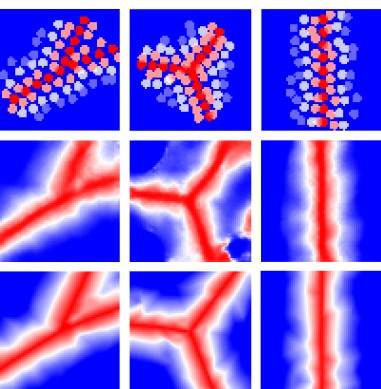

Figure 10: Sharpness field sampled from the *fandisk* model. Top row: sparse sampling of the sharpness field precomputed on the point cloud. Middle row: dense sharpness field predicted by our GAN. Bottom row: ground truth sharpness field obtained by casting rays onto the original mesh.

makes it simple to apply our GAN to locally predict the value of the sharpness field of a point cloud. If values of the sharpness field are pre-computed for the low-resolution point cloud, we can obtain a sparse image of the sharpness values simultaneously with the sparse heightmap (top row of Figure 10). By obtaining ground truth sharpness fields on meshes, we can then train a separate conditional GAN to predict the values of the sharpness field corresponding to all points of the dense heightmap (middle row of Figure 10).

In order to predict values of the sharpness field at the dense output points from our domain translation method, we trained a separate GAN to predict sharpness field values at the output points, using the same training procedure and data augmentation as the heightmap translation GAN. The only difference is that we obtained better results with a larger patch size of $16 \times 16$ for the discriminator. The training data is a set of meshes of simple geometric shapes, whose sharpness fields are computed using dihedral angles as described in [19]. We then pre-computed the sharpness field of the *blade* point cloud (80K points) using the CNN with a spatial transformer as recommended in [19], before performing domain translation using the *even* mode. The results we have obtained (Figure 11) show that our GAN gives a sharpness field with similar properties to a sharpness field computed from scratch on the dense point cloud after domain translation. Furthermore, the entire domain translation procedure along with the sharpness field estimation takes around 7 minutes, compared to 20 minutes if the sharpness field is recomputed from scratch on the dense point cloud. The combination of the dense heightmap, the normal map and the sharpness field together provide all the information necessary for downstream methods to accurately reconstruct the surface along with sharp features. We believe that our domain translation approach to upsampling has the potential for extension to other scalar fields and vector fields.

## 7 Conclusion, Limitations and Future Work

In this work, we have applied GAN-based domain translation to enable us to reconstruct fine features of low-resolution point clouds. We have obtained results superior to the state of the art, while providing a very different approach from previous PointNet++-based work. We have also given a simple example demonstrating that the same method has the potential to be extended to upsampling of scalar fields. Our work demonstrates that tangible benefits can be obtained by applying the domain translation paradigm to 3D geometry problems, and not merely the typical domains associated with GAN-based domain translation.

While our method is good at reconstructing low-level details, it

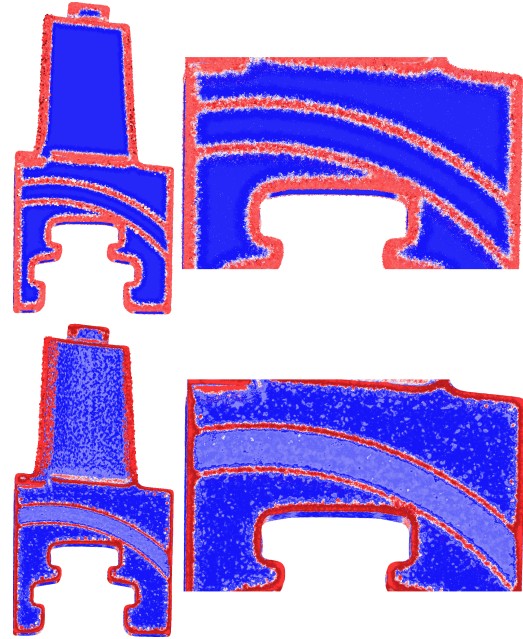

Figure 11: Upsampling results for the *blade* model, along with its sharpness field. Left: using our GAN to estimate the sharpness field. Right: recomputing the sharpness field from scratch on the dense point cloud.

has a slight tendency to add unnecessary detail in undersampled regions. Since we are performing domain translation, the GAN is forced to hallucinate a realistic-looking dense heightmap even in the presence of insufficient data. This problem opens up possibilities for future work.

In our problem, we are able to obtain paired training data from the two domains. The same approach can be extended to unpaired data in domains where it is difficult to align the data of multiple domains (e.g. heightmaps obtained from depth cameras and from laser scanners). The raycasting-based approach to generating ground truth data also opens up the possibility of obtaining training data from implicit or parametric surfaces such as CAD models, instead of meshes. There is also scope for extending our method to estimate other scalar or vector fields in tandem with upsampling.

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

## APPENDIX A   TRAINING DETAILS

### A.1   Generator architecture

The generator used in our GAN has a U-Net architecture with the following layers:

1. $4 \times 4$ convolution, stride 2, 64 kernels, leaky ReLU(slope= 0.2)

2. $4 \times 4$ convolution, stride 2, 128 kernels, instance norm., leaky ReLU(slope= 0.2)

3. $4 \times 4$ convolution, stride 2, 256 kernels, instance norm., leaky ReLU(slope= 0.2)

4. $4 \times 4$ convolution, stride 2, 512 kernels, instance norm., leaky ReLU(slope= 0.2), 50% dropout

5. $4 \times 4$ convolution, stride 2, 512 kernels, instance norm., leaky ReLU(slope= 0.2), 50% dropout

6. $4 \times 4$ convolution, stride 2, 512 kernels, leaky ReLU(slope= 0.2), 50% dropout

7. $4 \times 4$ transposed convolution, stride 2, 512 kernels, instance norm., ReLU, 50% dropout

   applied to concatenated output of layers 5 and 6.

8. $4 \times 4$ transposed convolution, stride 2, 512 kernels, instance norm., ReLU, 50% dropout

   applied to concatenated output of layers 4 and 7.

9. $4 \times 4$ transposed convolution, stride 2, 256 kernels, instance norm., ReLU

   applied to concatenated output of layers 3 and 8.

10. $4 \times 4$ transposed convolution, stride 2, 128 kernels, instance norm., ReLU

    applied to concatenated output of layers 2 and 9.

11. $4 \times 4$ transposed convolution, stride 2, 64 kernels, instance norm., ReLU

    applied to concatenated output of layers 1 and 10.

12. 2x upsampling

13. $4 \times 4$ convolution with bias, stride 1 with zero padding, 1 kernel, Tanh

## A.2  Discriminator architecture

Our discriminator uses the PatchGAN architecture proposed by Isola *et al*:

1. $4 \times 4$ convolution with bias, stride 2, 512 kernels, leaky ReLU(slope= 0.2)

2. $4 \times 4$ convolution with bias, stride 2, 1024 kernels, instance norm., leaky ReLU(slope= 0.2)

3. $4 \times 4$ convolution with bias, stride 2, 2048 kernels, instance norm., leaky ReLU(slope= 0.2)

4. $4 \times 4$ convolution, stride 1 with zero padding, 1 kernel, linear activation

We use a patch size of $8 \times 8$ for our heightmap domain translation experiments. For scalar field domain translation, we obtained better results with $16 \times 16$ patches, therefore layer 3 is omitted.

## A.3  Training hyperparameters

The following hyperparameters are identical for both the generator and the discriminator:

batch size $= 16$

optimizer: Adam

learning rate $= 3 \times 10^{-4}$

momentum: $\beta_1 = 0.5, \beta_2 = 0.999$

The discriminator loss is the mean squared error of classifying each patch as real or fake. The generator has to maximize the discriminator loss, while also minimizing the $\ell_1$ error of the predicted image. The $\ell_1$ error is given a weight of 10 for heightmap domain translation, and 0.5 in the case of scalar field domain translation.

## APPENDIX B   ADDITIONAL RESULTS

The following pages contain figures of additional results on testing shapes. Figures 12 and 13 show results obtained on inputs of 5000 sampled points. Figure 12 compares the reconstructed meshes, while Figure 13 shows the point clouds. As mentioned in Section 5 of the main paper, if the final objective is to consume a point cloud and not a mesh, our best results are obtained by either using the *even* mode, or by performing Poisson disk sampling on the mesh reconstructed using the *detailed* mode.

In Figure 14, we have also shown an example of how multiple random samplings of 625 points give slightly different reconstructed surfaces. Our results are compared with 3PU, which we found to produce less consistent results. We have omitted examples where PCA failed to give good enough normals for the 3PU output point cloud, resulting in failure of the screened Poisson reconstruction. Our method provides normals for output points, so no additional normal estimation step is required, and our computed normals never caused screened Poisson reconstruction to fail in our experiments.

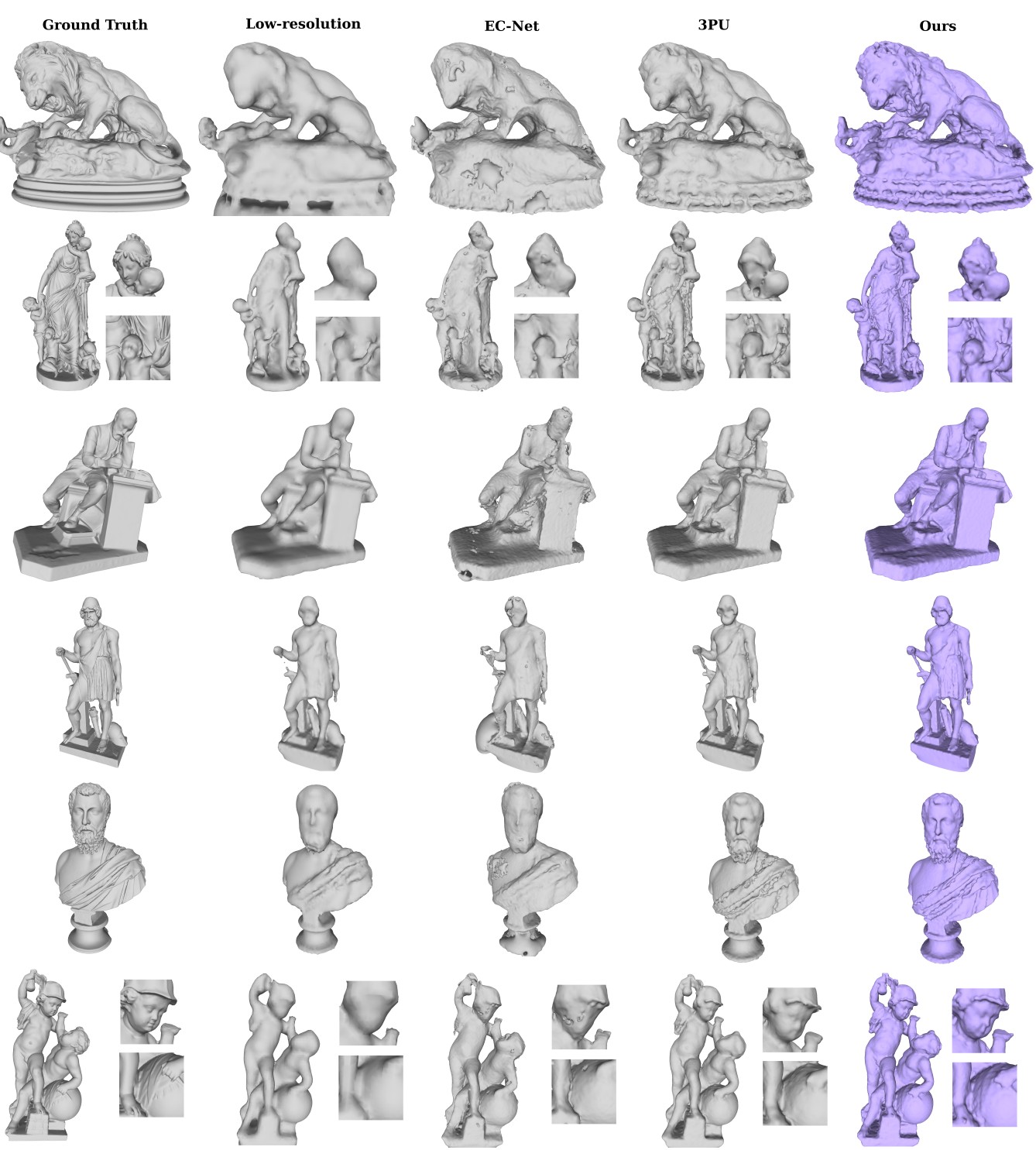

Figure 12: Additional results for surface reconstruction using our *detailed* mode domain translation, compared with 3PU and EC-Net.

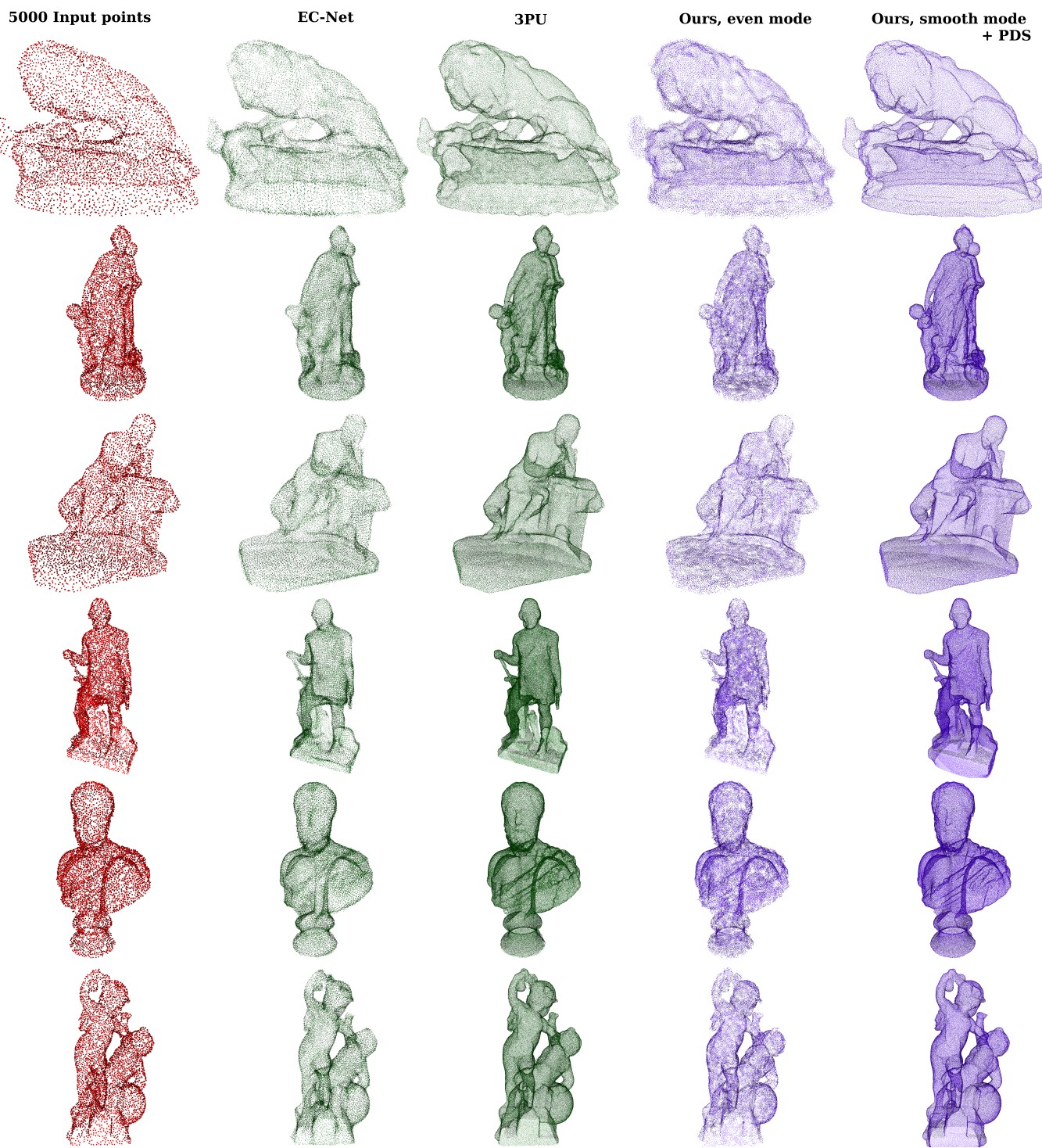

Figure 13: Dense point clouds obtained using domain translation using the *even* mode, compared with state-of-the-art point cloud upsampling methods. The last column shows results from Poisson disk sampling of our reconstructed surface, using the *detailed* mode.

| **625 Sampled Points** | **3PU** | **Ours** |
| --- | --- | --- |

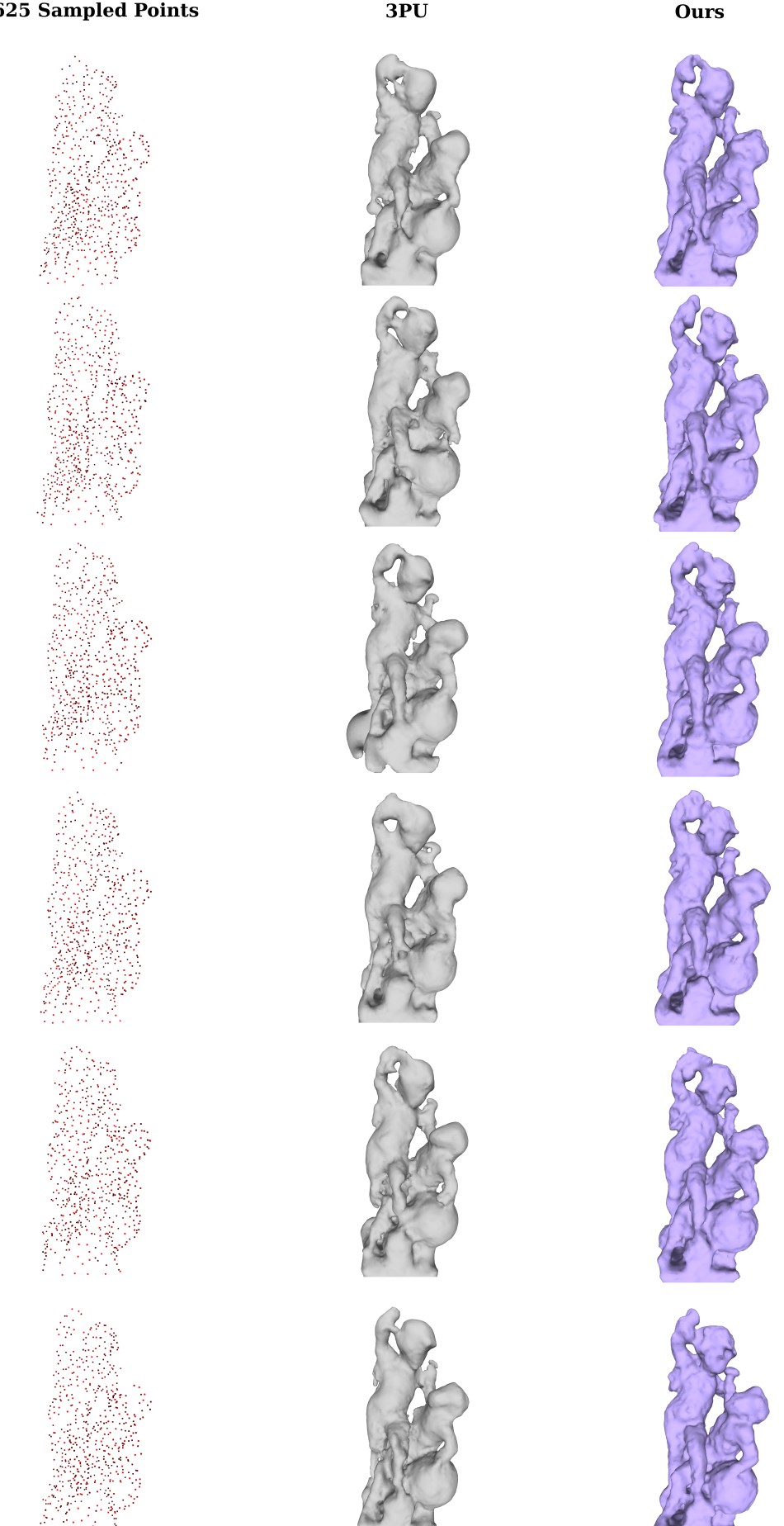

Figure 14: Comparison of surface reconstruction results obtained from multiple random samplings of 625 points of the *Cupid Fighting* model.