# OpenReview forum: "Fine Feature Reconstruction in Point Clouds by Adversarial Domain Translation"
_graphicsinterface.org/Graphics_Interface/2020/Conference — GI 2020_

### Official Review · AnonReviewer1 · 2020-01-06
**This paper addresses the point cloud upsampling problem by viewing it as an “domain translation” problem and using a condition.**

**Confidence:** 3
**Rating:** 4

**Review:**

This paper addresses the point cloud upsampling problem by viewing it as an “domain translation” problem and using a conditional GAN framework. More concretely, the proposed method first constructs sparse heightmaps from local patches of point clouds and feeds them as inputs to a generative adversary network to translate them into denser heightmaps. The resulted heightmaps are then compared with the ground-truth heightmaps, which are generated from the ground-truth meshes using raycasting. Finally, the upsampled point clouds can be obtained by reprojecting the 2D heightmaps into point clouds. The authors claimed that their results are superior to the state-of-the-art methods and their method is faster.

1)	It seems that the proposed method handles each patches of an input point cloud independently, does the number of these local patches constructed affect the final reconstruction result? How to determine this number?
2)	The training details of the compared methods EC-Net [25] and 3PU [19] are missing. Are they trained with the same dataset with the proposed method?
3)	More quantitative comparison results on common datasets, like ModelNet10 used in 3PU, should be provided.
4)	The proposed method is very similar with Pointpronets [17], except that it leverages a conditional GAN architecture. An ablation study should be conducted to show its necessity and effectiveness.
5)	It seems that the comparison is based on the reconstructed meshes using the screened Poisson surface reconstruction, which requires the vertex normals as input. While the proposed method also predicts vertex normal and uses them in surface reconstruction, the vertex normal of the compared methods are estimated in a relatively simple way. This may cause the comparison unfair. Since the main contribution of the paper is on upsampling of point clouds, the direct comparison of the output point clouds should be provided.
6)	According to the authors, “different random samplings of 625 input points for the same testing mesh give slightly different resulting meshes”. How did the numbers in quantitative comparison calculated? Are they average value of several different runs or you just chose the best one among them?

---

### Official Review · AnonReviewer2 · 2020-01-06
**A domain translation technique for upsampling point cloud**

**Confidence:** 3
**Rating:** 6

**Review:**

This paper presents a data-driven point cloud upsampling method using conditional GANs. The key idea is to process local oriented point cloud patches, instead of the entire point cloud, and represent the local patch as a height map image in order to leverage the power of image-based network architectures. Such technique generalizes well to low-res input point clouds and shows superior performance comparing to previous works.

Overall, the results are convincing and the network seems to generalize well, but I still have some concerns and questions.

1. How does the approach performs on real-world scanned point clouds? It seems like the training data comes from uniformly sampled (Poisson disk) points on meshes. I wonder how does the approach generalize to scanned point clouds with missing regions due to occlusions.

2. How is the orientation of each height-map image determined?

3. A follow-up question is how does the overlapped region between adjacent patches be handled? Two adjacent height-map images may result in different predictions within the overlapped region, due to inconsistent orientations or the network. I wonder how does the approach handle such inconsistency.

In short, the results are convincing. The ability to generate details from low-resolution point cloud is promising. My main concern is on the applicability of the method to real-world scenarios because this method seems requiring the input to be a uniformly sampled oriented point cloud.

---

### Official Review · AnonReviewer3 · 2020-01-07
**Upsampling pointclouds via a GAN on local height maps**

**Confidence:** 3
**Rating:** 6

**Review:**

### Summary

This submission presents a method for upsampling point clouds using a GAN. The method operates on height maps over local patches of a sparse point cloud; for each height map from the sparse cloud, the GAN outputs a corresponding dense map, which can then be sampled to yield a dense cloud. The GAN is trained by sampling heightmaps from a collection of high-quality meshes, as well as downsampled versions of those same meshes.

The method yields visually plausible results, and is shown to be competitive-with or outperform recent work according to several evaluation criteria. The submission also demonstrates applications to upsampling extremely sparse point clouds, and upsampling scalar fields.

### Feedback

Overall, the method seems sound, and the presentation is acceptable. The basic idea of performing upsampling in height-map space complements other neighborhood-based strategy, and it is makes sense that this allows simple and powerful image-based networks to be used.

The presentation of the method as "domain translation using GANs" strikes me as odd, because it is not really the GAN that is doing domain translation---the GAN operates exclusively on images. It is the raycaster/rasterizer which translate from mesh/point cloud to image/heightmap domains. Is this terminology typical?

The experimental evidence is barely-adequate. Although the submission compares to two recent methods, the comparison is only on a small, manually collected dataset presented in this work.

At least one recent work is unmentioned, which also uses GANs to perform upsampling (though the rest of the methodology is quite different):

> PU-GAN: a Point Cloud Upsampling Adversarial Network. (ICCV 2019)

Given how recent this competing work is, I'm okay with not having a comparison, but it should be cited.

### Questions:

- Section 4, paragraph 2 indicates that normals for the sparse cloud are computed using 30-neighbor PCA. Are these neighbors in the sparse cloud, or the original dense cloud? If it is the dense cloud, this would seem to be propagating information from the ground truth via the normals.

- The last paragraph of section 4.1 is not clear to me. When the text says "This variation is much larger...", which method does "this" refer to? Does the methodology perform multiple runs of this method, 3PU, or both?

- I'm surprised by the argument that PointNet++-based methods are not permutation-invariant (Section 5, list point 1). Choosing a random initial point yields a permutation invariant distribution of outputs (though this is not deterministic), and choosing the point nearest the cloud centroid gives full permutation invariance. This seems like a very small detail as a criticism of all PointNet++-based methods.

- The dataset sounds very similar to the Sketchfab dataset used in [19]. Is it the same? If so this is an important detail.

---

### Meta-Review · Area_Chair1 · 2020-01-11

**Recommendation:** Accept
**Confidence:** 3

**Metareview:**

It is recommended that the paper be accepted for presentation at GI'20. The merits of the submission include the importance of the topic and convincing results.  But the paper needs to be revised to improve its validation in terms of using a larger data sets and comparing with (or citing) other related works.  Please see the detailed review comments to improve the presentation of the paper.

---

### Decision · Program_Chairs · 2020-01-11

Accept